# Hyperbolic Busemann Learning with Ideal Prototypes

**Mina Ghadimi Atigh**
University of Amsterdam

**Martin Keller-Ressel**
Technische Universität Dresden

**Pascal Mettes**
University of Amsterdam

## Abstract

Hyperbolic space has become a popular choice of manifold for representation learning of various datatypes from tree-like structures and text to graphs. Building on the success of deep learning with prototypes in Euclidean and hyperspherical spaces, a few recent works have proposed hyperbolic prototypes for classification. Such approaches enable effective learning in low-dimensional output spaces and can exploit hierarchical relations amongst classes, but require privileged information about class labels to position the hyperbolic prototypes. In this work, we propose Hyperbolic Busemann Learning. The main idea behind our approach is to position prototypes on the ideal boundary of the Poincaré ball, which does not require prior label knowledge. To be able to compute proximities to ideal prototypes, we introduce the penalised Busemann loss. We provide theory supporting the use of ideal prototypes and the proposed loss by proving its equivalence to logistic regression in the one-dimensional case. Empirically, we show that our approach provides a natural interpretation of classification confidence, while outperforming recent hyperspherical and hyperbolic prototype approaches.

## 1 Introduction

Classification by prototypes has a long tradition in machine learning. Foundational solutions such as the Nearest Mean Classifier [46] represent classes as the mean prototypes in a fixed feature space. Following approaches that learning a metric space for class prototypes [28], deep learning with prototypes as points in network output spaces has gained traction [14, 17, 26, 32, 36, 41, 49, 51]. In line with foundational approaches, the prototypes are positioned by computing the mean over all training examples for each class. Deep learning by mean prototypes has shown to be effective for tasks such as few-shot learning [14, 32, 36, 41] and zero-shot recognition [41, 49, 51].

To avoid the need to re-learn prototypes or to enable the use of prior label knowledge, several recent works have proposed deep networks with prototypes in non-Euclidean output spaces. Hyperspherical prototype approaches initialize prototypes on the sphere or its higher-dimensional generalization, based on pair-wise angular separation [29, 39] or with the help of word embeddings [4, 43] using a cosine similarity loss between example outputs and class prototypes. Hyperspherical prototypes alleviate the need to re-position prototypes continuously and allow to optionally include prior semantic knowledge about the classes. Recently, a few works have extended prototype-based learning to the hyperbolic domain, where prototypes are obtained by embedding label hierarchies [23, 24]. While hyperbolic prototype approaches provide more hierarchically coherent classification results, prior hierarchical knowledge is required to embed prototypes. This paper strives to combine the best of both non-Euclidean worlds, namely efficient low-dimensional embeddings from hyperbolic prototypes and the knowledge-free positioning from hyperspherical prototypes.

We make three contributions in this work. First, we introduce a hyperbolic prototype network with class prototypes given as points on the ideal boundary of the Poincaré ball model of hyperbolic geometry. Second, we propose the penalized Busemann loss, which enables us to compute proximities between example outputs in hyperbolic space and prototypes at the ideal boundary, an impossible task for existing distance metrics, which put the ideal boundary at infinite distance from all other

35th Conference on Neural Information Processing Systems (NeurIPS 2021)

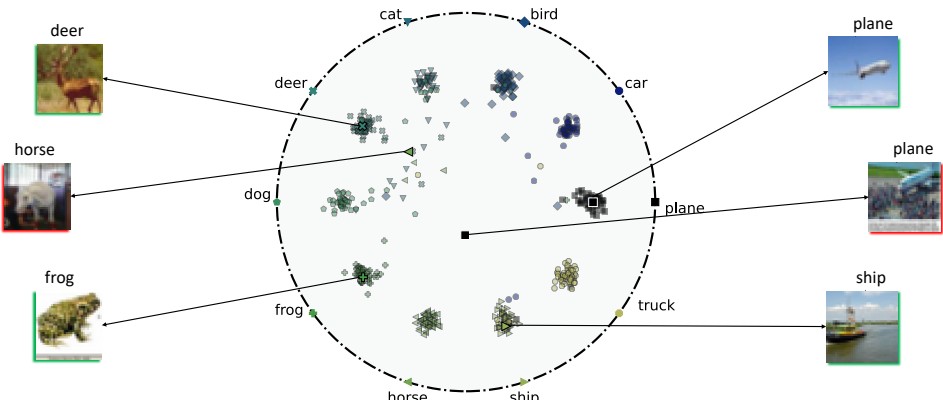

Figure 1: **Visualization of the output space for Hyperbolic Busemann Learning** in two dimensions on CIFAR-10. The penalized Busemann loss optimizes images to be close to their corresponding ideal prototypes while forcing all examples towards the origin to avoid over-confidence, *e.g.,* for the *deer*, top *plane*, and *ship* examples. Ambiguous cases, such as the *horse* and bottom *plane* examples are projected closer to the origin, providing a natural interpretation of classification confidence.

points in hyperbolic space. Third, we provide a theoretical link between our hyperbolic prototype approach and logistic regression. We show that our choices for output manifold, for prototypes on the ideal boundary, and for the proposed loss provide a direct generalization of the logistic regression model. Experiments on three datasets show that our approach outperforms both hyperspherical and hyperbolic prototype approaches. Moreover, our hyperbolic output space provides a natural interpretation of closeness to the ideal boundary as classification confidence, see Figure 1. The code is available at https://github.com/MinaGhadimiAtigh/Hyperbolic-Busemann-Learning.

## 2 Related work

As an early adaptation of prototypes in deep networks, the center loss [47] served as an additional loss for softmax cross-entropy optimization, where all outputs of a class were pulled to one point to help minimize intra-class variation. Closer to the original Nearest Mean Classifier objective, Prototypical Networks introduce a prototype-only approach where each class is represented as the mean output of its corresponding training examples [41]. Learning with prototypes from class means has shown to be effective for few-shot learning [14, 32, 36, 41], zero-shot recognition [41, 49, 51], and domain adaptation [33]. Defining prototypes as class means is feasible in few-shot learning by sampling all samples of a class in a mini-batch. For general supervised learning, DeepNCM [17] and DeepNNO [26] propose alternating optimization schemes, where the prototypes are fixed to update the network backbone and vice versa. To avoid the need for constant prototype updating, several works have proposed to use the hypersphere as output space instead of the Euclidean space. On the hypersphere, prototypes can be positioned *a priori* based on angular separation [29, 39], using an orthonormal basis [4], or using prior knowledge from word embeddings [4, 29, 43]. Similar to hyperspherical prototype approaches, we seek to position prototypes at unit distance from the origin. In contrast, we operate on a hyperbolic manifold, which provides a theoretical link to logistic regression and obtains better empirical results than a hypershperical manifold and loss.

In this work, we strive for a prototype-based network on hyperbolic manifolds, which have become a popular choice for embedding data and deep network layers. Hyperbolic embeddings have shown to represent tree-like structures [38, 37], text [1, 44, 55], cells [20], networks [18], multi-relational knowledge graphs [3], and general taxonomies [16, 21, 30, 31, 50] with minimal distortion. In deep networks, hyperbolic alternatives have been proposed for fully-connected layers [15, 40], convolutional layers [40], recurrent layers [15], classification layers [13, 15, 40], and optimizers [5, 6, 52]. Hyperbolic generalizations have also been introduced for graph networks [2, 12, 22, 25, 53, 54] and generative models [7, 27]. Where hyperbolic-based deep learning approaches commonly operate within the Poincaré model represented by the open unit ball, we propose to position class prototypes

on its ideal boundary – the shell of the unit ball – allowing us to benefit from the representational power of hyperbolic geometry without requiring prior knowledge to operate.

A few works have recently proposed prototype-based approaches with hyperbolic output spaces. Hyperbolic Image Embeddings by Khrulkov *et al.* [19] include a prototype-based setting as a direct hyperbolic generalization of Prototypical Networks [41], allowing for few-shot learning through hyperbolic averaging. Beyond few-shot learning, Long *et al.* [24] introduce a hyperbolic action network, where action prototypes are projected to a hyperbole based on prior knowledge about hierarchical relations. Videos are in turn projected to the same space and compared to the prototypes using the hyperbolic distance, enabling hierarchically consistent action recognition and zero-shot recognition. Similarly, Liu *et al.* [23] perform zero-shot recognition using hyperbolic prototypes positioned based on known hierarchical relations. In this work, we also employ prototypes in hyperbolic space, but do not require prior knowledge about hierarchies. Recent approaches with hyperbolic output spaces have shown to be effective in low dimensionalities [19, 24, 23]. Low dimensional embeddings have several practical use cases and benefits. We list two important use cases. *Compactness:* Hyperbolic Busemann Learning enables us to embed high-dimensional inputs in low-dimensional output spaces. This ability has potential benefits in applications such as data transmission, compression, and storage. *Interpretability:* Obtaining high performance with few output dimensions (*i.e.,* two or three output dimensions) provides the potential to interpret the model performance and visualize the output space, see Figure 1.

## 3 Learning with hyperbolic ideal prototypes

### 3.1 Problem formulation

We are given a training set $\{(\mathbf{x}_i, y_i)\}_{i=1}^N$ with $N$ examples, where $\mathbf{x}_i \in \mathbb{R}^I$ denotes an input with dimensionality $I$ and $y_i \in \{1, .., C\}$ denotes a label from a set of size $C$. We seek to learn a projection of inputs to a hyperbolic manifold in which can compute proximities to class prototypes. We do so through a transformation $\mathcal{F}$ in Euclidean space, followed by an exponential map from the tangent space $\mathcal{T}_x \mathcal{M}$ to a hyperbolic manifold $\mathcal{M}$:

$$\mathbf{z} = \exp_{\mathbf{v}}(\mathcal{F}(\mathbf{x}; \theta)), \tag{1}$$

where $\theta$ denotes the parameters to learn the transformation. The transformation can be any function, e.g., a deep network or a linear layer akin to logistic regression. The hyperbolic manifold in which $\mathbf{z}$ operates is described throughout this work by the Poincaré ball model $(\mathbb{B}_d, g_{\mathbf{z}}^{\mathbb{B}})$, with open ball $\mathbb{B}_d = \{\mathbf{z} \in \mathbb{R}^d \mid ||\mathbf{z}||^2 < 1\}$ and the Riemannian metric tensor $g_{\mathbf{z}}^{\mathbb{B}} = 4(1 - ||\mathbf{z}||^2)^{-2} \mathbf{I}_d$ [11, 34]. Setting its center to $\mathbf{v} = 0$ the exponential map in the Poincaré model is given by:

$$\exp_0(\mathbf{x}) = \tanh(||\mathbf{x}||/2) \frac{\mathbf{x}}{||\mathbf{x}||}. \tag{2}$$

Furthermore, the geodesic distance between two points $z_1, z_2 \in \mathbb{B}_d$ is given by:

$$d_{\mathbb{B}}(\mathbf{z}_1, \mathbf{z}_2) = \text{arcosh}\left(1 + 2\frac{||\mathbf{z}_1 - \mathbf{z}_2||^2}{(1 - ||\mathbf{z}_1||^2)(1 - ||\mathbf{z}_2||^2)}\right). \tag{3}$$

To train our model, we compare the projected inputs to their class labels, represented by the prototypes $P = \{\mathbf{p}_1, .., \mathbf{p}_C\}$. That is, the overall training loss is given as the aggregation of distance-based individual losses $\ell$ between projected inputs and the prototype of their respective class labels:

$$\mathcal{L}(\theta) = \frac{1}{N} \sum_{i=1}^N \ell(\exp_0(\mathcal{F}(\mathbf{x}; \theta)), \mathbf{p}_{y_i}). \tag{4}$$

### 3.2 Ideal prototypes and the penalized Busemann loss

The central idea of our approach is to position class prototypes at ideal points, which represent points at infinity in hyperbolic geometry. In the Poincaré model, the ideal points form the boundary of the ball:

$$\mathbb{I}_d = \{\mathbf{z} \in \mathbb{R}^d : z_1^2 + \cdots + z_d^2 = 1\}. \tag{5}$$

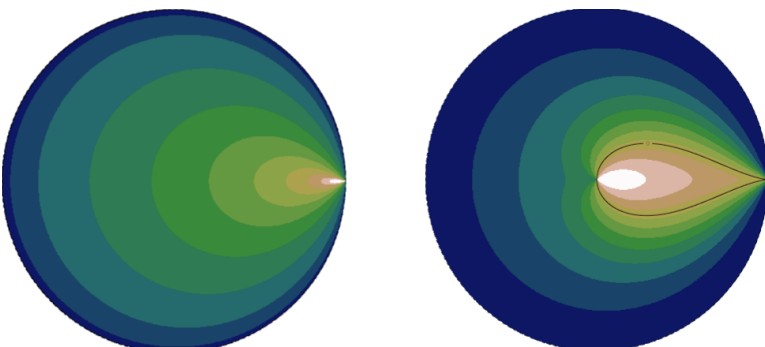

Figure 2: **Visualization of the penalized Busemann loss.** The left plot shows a color map of the penalized Busemann loss function, see (8), in dimension two for a single ideal prototype located at $\mathbf{p} = (1, 0)$; contour lines are log-spaced. The right plot shows its gradient with respect to the radial coordinate. During loss minimization, points in the drop-shaped region adjacent to the prototype will move outwards, while points outside of the region will move inwards.

Thus, the set of ideal points of hyperbolic space $\mathbb{B}_d$ is homeomorphic to the hypersphere $\mathbb{S}_d$. As a consequence, any of the methods used for prototype embedding into $\mathbb{S}_d$ can be used to embed prototypes into $\mathbb{I}_d$. In particular, for $d = 2$, prototypes can be placed uniformly on the unit sphere $\mathbb{S}_2$, while class-agnostic prototype embedding based on separation can be used for $d \geq 3$ [29].

A key issue that needs to be solved, is that ideal points are at infinite geodesic distance from all other points in $\mathbb{B}_d$ and hence the geodesic distance can not be used directly for prototype-based learning. Therefore, we propose the penalized Busemann loss for hyperbolic learning with ideal prototypes. The Busemann function, originally introduced in [9] (see also Def. II.8.17 in [8]), can be considered a distance to infinity and may be defined in any metric space. Let $\mathbf{p}$ be an ideal point and $\gamma_{\mathbf{p}}$ a geodesic ray, parametrized by arc length, tending to $\mathbf{p}$. Then the Busemann function with respect to $\mathbf{p}$ is defined for $\mathbf{z} \in \mathbb{B}_d$ as:

$$b_{\mathbf{p}}(\mathbf{z}) = \lim_{t \to \infty} (d_{\mathbb{B}}(\gamma_{\mathbf{p}}(t), \mathbf{z}) - t). \tag{6}$$

In the Poincaré model, the limit can be explicitly calculated and the Busemann function is given as:

$$b_{\mathbf{p}}(\mathbf{z}) = \log \frac{||\mathbf{p} - \mathbf{z}||^2}{(1 - ||\mathbf{z}||^2)}. \tag{7}$$

The full derivation of (7) is provided in Appendix A. Our proposed loss combines the Busemann function with a penalty term:

$$\ell(\mathbf{z}, \mathbf{p}) = b_{\mathbf{p}}(\mathbf{z}) - \phi(d) \cdot \log(1 - ||\mathbf{z}||^2), \tag{8}$$

with $\phi(d)$ a scaling factor for the penalty term which is a function of the dimension of the hyperbolic space. The first term in the penalized Busemann loss steers a projected input $\mathbf{z}$ towards class prototype $\mathbf{p}$, while the second term penalized overconfidence, *i.e.,* values close to the ideal boundary of $\mathbb{B}_d$. The prototypes are positioned before training and remain fixed, hence the backpropagation only needs to be done with respect to the inputs. The gradient of $\ell(\exp_0(\mathbf{x}), \mathbf{p})$ with respect to $\mathbf{x}$ is given as:

$$\nabla_{\mathbf{x}} \ell(\exp_0(\mathbf{x}), \mathbf{p}) = (\mathbf{x} - \mathbf{p}) \frac{\tanh(||\mathbf{x}||)}{||\mathbf{x}|| - \tanh(||\mathbf{x}||) \, \mathbf{p} \cdot \mathbf{x}} + \mathbf{1}_d \, \mathbf{p} \cdot \mathbf{x} \frac{\tanh(||\mathbf{x}||)/||\mathbf{x}|| - 1}{||\mathbf{x}|| - \tanh(||\mathbf{x}||) \, \mathbf{p} \cdot \mathbf{x}} + \tanh(||\mathbf{x}||/2), \tag{9}$$

with $\mathbf{1}_d$ a $d$-dimensional vector of ones. Figure 2 shows a color map of the penalized Busemann loss itself and its radial gradient for a prototype located at ideal point $\mathbf{p} = (1, 0)$.

In the penalized Busemann loss, $\phi(d)$ governs the amount of regularization. Below, we show that this function should be linear in the number of dimensions.

**Theorem 1.** *For any $d \geq 4$ and ideal point $\mathbf{p}$ the penalized Busemann loss $\ell(\mathbf{z}, \mathbf{p})$ is the log-likelihood of a density $f(\mathbf{z}, \mathbf{p}) = \frac{1}{C(d)} \exp\left(-\ell(\mathbf{z}, \mathbf{p})\right)$ on the Poincaré ball $\mathbb{B}_d$ if and only if $\phi(d) > d - 2$.*

*Proof.* To show that $f(\mathbf{z}, \mathbf{p})$ is a density, we have to show that the normalization constant

$$C(d) = \int_{\mathbb{B}_d} \exp\left(-\ell(\mathbf{z}, \mathbf{p})\right) dV$$

is finite. Here, $dV$ is the hyperbolic volume element on $\mathbb{B}_d$, which is equal to $2^d(1 - ||\mathbf{z}||^2)^{-d}dz_1 \cdots dz_d$, see [34]. Furthermore, using invariance of the integrand under rotation, we can assume without loss of generality that $\mathbf{p}$ is equal to the unit vector $e_1 = (1, 0, \ldots, 0)$, which leads to

$$C(d) = 2^d \int_{\mathbb{B}_d} \left(\frac{(1 - ||\mathbf{z}||^2)^{\phi(d)+1-d}}{1 - 2z_1 + ||\mathbf{z}||^2}\right) dz_1 \cdots dz_d.$$

Switching to (Euclidean) hyperspherical coordinates $(r, \varphi_1, \ldots, \varphi_d)$, the integral factorizes into $C(d) = 2^d \cdot C_1(d) \cdot C_2(d)$, where

$$C_1(d) = \int_0^1 \int_0^\pi \left(\frac{(1 - r^2)^{\phi(d)+1-d}}{1 - 2r\cos(\varphi_1) + r^2}\right) r^{d-1} \sin(\varphi_1)^{d-2} dr\, d\varphi_1$$

$$C_2(d) = \int_0^\pi \sin(\varphi_2)^{d-3} d\varphi_2 \ldots \int_0^\pi \sin(\varphi_{d-2})^2 d\varphi_{d-2} \cdot \int_0^{2\pi} \sin(\varphi_{d-1}) d\varphi_{d-1}.$$

Clearly, $C_2(d)$ is finite, independently of $d$, such that it remains to consider $C_1(d)$. Instead of solving the integral exactly, we use the elementary estimate

$$\sin(\varphi_1)^2 \leq 1 - 2r\cos(\varphi_1) + r^2 \leq 4,$$

valid for all $r \in [0, 1]$. This estimate allows us to obtain both a lower and an upper bound on $C_1(d)$. The upper bound becomes

$$C_1(d) \leq \int_0^1 (1 - r^2)^{\phi(d)+1-d}\, r^{d-1} dr \int_0^\pi \sin(\varphi_1)^{d-4} d\varphi_1,$$

which (given that $d \geq 4$) is finite if and only if $\phi(d) > d - 2$. The lower bound becomes

$$C_1(d) \geq \frac{1}{4} \int_0^1 (1 - r^2)^{\phi(d)+1-d}\, r^{d-1} dr \int_0^\pi \sin(\varphi_1)^{d-2} d\varphi_1,$$

which (given that $d \geq 2$) is also finite if and only if $\phi(d) > d - 2$, which completes the proof. $\square$

Given a trained model, inference is performed by projecting the test example $\mathbf{x}$ and selecting the class with the smallest prototype distance:

$$y^\star = \arg\min_{\mathbf{p} \in P} \ell(\exp_0(\mathcal{F}(\mathbf{x}; \theta)), \mathbf{p}). \tag{10}$$

The Poincaré ball model of hyperbolic geometry has the conformal property, *i.e.,* angles in the Poincaré ball model coincide with angles in Euclidean space [35]. Thus, the cosine similarity between two ideal points remains a meaningful measure of their similarity, even in hyperbolic geometry. As the penalty term is the same for all prototypes, we can equivalently minimize the Busemann function of the prototypes directly. Moreover, as the Busemann function is a decreasing function of the cosine similarity, prediction is equivalent to hyperspherical prototype inference:

$$y^\star = \arg\max_{\mathbf{p} \in P} \frac{\mathbf{z}}{||\mathbf{z}||} \cdot \mathbf{p}, \qquad \mathbf{z} = \exp_0(\mathcal{F}(\mathbf{x}; \theta)). \tag{11}$$

### 3.3   Relation to logistic regression

Logistic regression is a foundational algorithm for classification and a pillar in supervised learning for deep networks with cross entropy optimization. Here, we show that hyperbolic ideal prototype learning is a natural generalization of logistic regression by showing its equivalence when using $d = 1$ dimensions in our approach.

**Theorem 2.** *For dimension d = 1 and with penalty $\phi(1) = 1$, Hyperbolic Busemann Learning with a single linear layer is equivalent to logistic regression.*

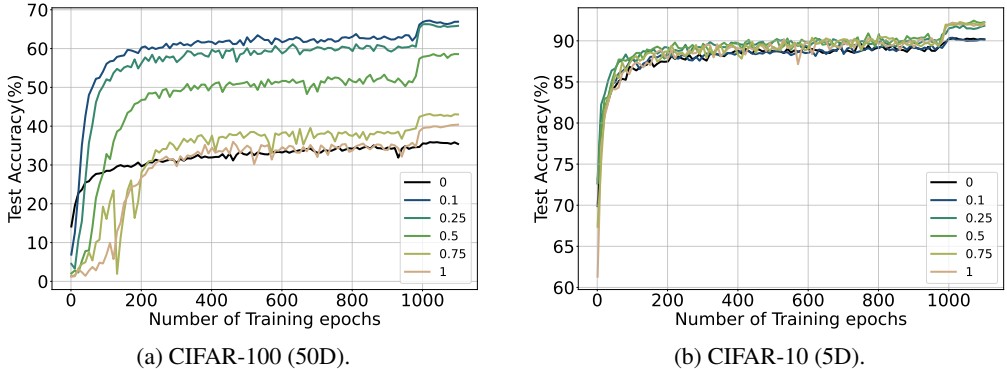

(a) CIFAR-100 (50D).          (b) CIFAR-10 (5D).

Figure 3: **Ablation study on the penalized Busemann loss** for CIFAR-100 and CIFAR-10. On both datasets, adding a non-negative penalty (0.1 to 1) improves the classification results over the Busemann loss without penalty (0). With more classes, a lower penalty is preferred. The y-axis on CIFAR-10 starts from 60% to better visualize differences between settings.

*Proof.* Recall that from the machine learning point of view logistic regression can be described as a single linear layer in combination with the logistic activation function and cross-entropy-loss. In one dimension, hyperbolic space $\mathbb{B}_1$ becomes the interval $(-1, 1)$. The set of ideal points $\mathbb{I}_1$ consists of the two points $\pm 1$. To identify these prototypes with $0/1$ (the class labels in logistic regression) we introduce a linear change of coordinates to

$$z' = \frac{z+1}{2}.$$

Under this change of coordinates $\mathbb{B}_1$ becomes the unit interval $(0, 1)$ and the prototypes $p'$ (the ideal points) become the endpoints $0$ and $1$. The exponential map, i.e., the embedding of the base learner output $y = w^\top x + w_0$ into $\mathbb{B}_1$, maps $y \in \mathbb{R}$ to

$$z' = \exp_0(y) = \frac{\tanh(y/2) + 1}{2} = \frac{1}{1 + e^{-y}},$$

which is the logistic function. In $z'$-coordinates, the penalized Busemann loss becomes

$$\ell(z', p') = 2\log\left(\frac{|p' - z'|}{2z'(1 - z')}\right) = \begin{cases} 2\log\left(\frac{1}{2(1-z')}\right) & \text{if } p' = 0 \\ \log\left(\frac{1}{2z'}\right) & \text{if } p' = 1 \end{cases}.$$

Rewriting this as

$$\frac{1}{2}\ell(z', p') + \log(2) = -p'\log(z') - (1 - p')\log(1 - z'),$$

shows that the shifted and scaled penalized Busemann loss coincides with the cross-entropy loss in dimension $d = 1$. □

Beyond binary classification, multinomial logistic regression is performed in a one-vs-rest manner with a $K$-dimensional softmax output for $K$ classes. *i.e.,* in conventional multinomial logistic regression the output space scales one-to-one with the number of classes. With Hyperbolic Busemann Learning, we provide a different generalization of logistic regression, where the number of output dimensions is independent of the number of classes. Most prominently, we find that most of the classification performance can be maintained in output spaces of small dimension, cf., Tables 1 and 4.

## 4  Experiments

### 4.1  Ablation studies

**Setup.** We first investigate the workings of the penalized Busemann loss. We evaluate the effect of the penalty term on CIFAR-10 and CIFAR-100 using a ResNet-32 backbone. For the experiment, we

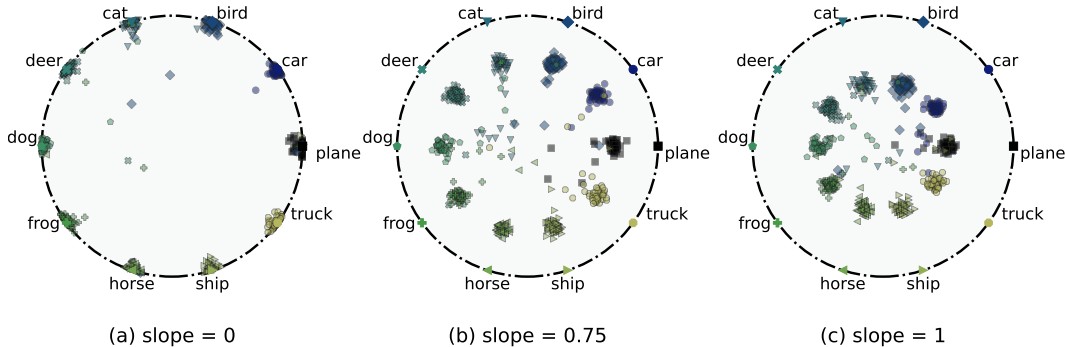

| (a) slope = 0 | (b) slope = 0.75 | (c) slope = 1 |

Figure 4: **Visualization of the penalty term** and its effect on the output space with a ResNet-32 on CIFAR-10. Without a penalty, examples are close to the boundary, leading to overconfidence. Contrarily, a high penalty forces examples towards the origin of the Poincaré ball, leading to inter-class confusion. With the right slope, these factors are balanced, resulting in improved classification.

use Adam with a learning rate of 5e-4, weight decay of 5e-5, batch size of 128, without pre-training. The network is trained for 1,110 epochs with learning rate decay of 10 after 1,000 and 1,100 epochs. The prototypes are given by [29] without prior knowledge.

**Results.** In Figure 3, we show the results for CIFAR-100 and CIFAR-10 for various settings of the penalty term in the Busemann loss. The penalty term $\phi(d)$ depends on the output dimensionality and should be linear in form as per Theorem 1. In the Figure, we show five settings, corresponding to the slope $s$ of the linear function, *i.e.,* $\phi(d; s) = s \cdot d$. On both datasets the setting with $s=0$, where the penalty term is excluded, performs worst, highlighting the importance of including a penalty in the Busemann loss. For the other settings, the higher the value, the larger the penalty and the stronger the pull towards the origin of the output space. On CIFAR-10, all non-zero slopes improve the classification accuracy. Slope $s=0.75$ is slightly preferred with an accuracy of 92.3% versus 91.9% for $s=1$ and 90.1% for $s=0$. For CIFAR-100, a too strong penalty negatively impacts the classification results. A penalty of $s=0.1$ performs best with an accuracy of 65.8% compared to 39.7% and 35.9% for $s=1$ and $s=0$. We conclude that the penalty term is an influential component of the proposed loss and with more classes a lower slope is recommended for best performance.

**Analysis.** To better understand the penalty term and its effect, we have trained a ResNet-32 backbone on CIFAR-10 using only two output dimensions. In Figure 4, we visualize the output space for three slope values, along with the ideal prototypes and test examples. The visualization clearly shows the importance of the penalty; without it, examples move closely towards the ideal prototypes leading to over-confidence. On the other hand, a high penalty pulls all examples close to the origin of the Poincaré ball. Especially with many classes, this leads to smaller inter-class variation and higher confusion. The visualization explains the lower performance for a high penalty on CIFAR-100 compared to CIFAR-10. By balancing confidence and inter-class separation, the best classification results are achieved. Figure 1 and Figure 4 show intuitively that the distance to the origin for an example reflects the confidence of the network in classifying the example. We have performed a quantitative analysis to show that our approach provides a natural interpretation of classification confidence, see Figure 5. In general, the closer the example to the origin, the lower the overall confidence, as also indicated in other hyperbolic approaches [19, 42]. The red line in Figure 5 shows that the distance to the origin in our approach also correlates with angular similarity to the correct class prototype. In other words, the larger the distance to the origin, the higher the likelihood of the classification being correct. Again, this provides a parallel to logistic regression, where the

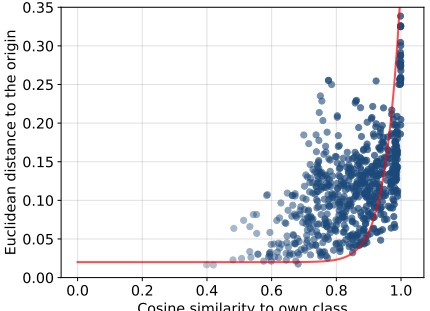

Figure 5: **Confidence interpretation.** The larger the distance to the origin, the more likely the example is correctly classified, showing that our hyperbolic approach provides a natural interpretation of classification confidence.

Table 1: **Comparison to Hyperspherical Prototype Networks** on CIFAR-100 and CIFAR-10 using the same prototypes and a ResNet-32. * denotes results from code provided by [29] in case the number was not reported in the original paper. For both datasets, our approach obtains a high accuracy, especially with few output dimensions.

| | CIFAR-100 | | | | | CIFAR-10 | | | |
|---|---|---|---|---|---|---|---|---|---|
| *Dimensions* | 3 | 5 | 10 | 25 | 50 | 2 | 3 | 4 | 5 |
| Mettes *et al.* [29] | 5.5 | 28.7 | 51.1 | 63.0 | 64.7 | 30.0* | 84.9* | 88.8* | 89.5* |
| This paper | **49.0** | **54.6** | **59.1** | **65.7** | **65.8** | **91.2** | **92.2** | **92.2** | **92.3** |

output score can be interpreted as a classification probability, conditioned on the input. We conclude from the analysis that the severity of the linear penalty matters to balance over-confidence and inter-class confusion, while the networks outputs enable a direct interpretation of the classification results.

## 4.2 Comparison to hyperspherical prototypes

**Setup.** In the second experiment, we compare our approach to its hyperspherical counterpart, in which the output space is the positively curved hypersphere instead of negatively curved hyperbolic space. We draw a comparison to Hyperspherical Prototype Networks [29], which has shown to be competitive or better than conventional softmax cross-entropy and using Euclidean prototypes. For a fair comparison, we use the same prototypes and the same network backbone. The difference lies in the choice of manifold and accompanying loss function. Following [29], we report on CIFAR-100, CIFAR-10, and CUB Birds 200-2011 [45] across multiple dimensionalities. On CIFAR-100, we use 3, 5, 10, 25, and 50 output dimensions, while we use 2, 3, 4, and 5 output dimensions for CIFAR-10 and 3, 10, 50, and 100 for CUB Birds 200-2011. For CUB Birds 200-2011, we keep the main hyperparameters the same as for the CIFAR experiments and use a weight decay of 0.0001, 2110 epochs, and a slope of 0.05.

**Results.** We show the classification results for CIFAR-100 and CIFAR-10 on ResNet-32 in Table 1 and CUB Birds 200-2011 in Table 2. Across all dimensions, our approach performs better and the gap increases when using fewer dimensions. On CIFAR-100 for five output dimensions, we obtain 54.6% accuracy versus 28.7% for hyperspherical prototypes and the relative difference grows for three dimensions with 49.0% versus 5.5%. On CUB Birds 200-2011 for ten output dimensions, we obtain an accuracy of 43.5% versus 9.5% for hyperspherical prototypes. When scaling down to three dimensions,

Table 2: **Comparison to Hyperspherical Prototype Networks** on CUB Birds 200-2011 using the same prototypes and a ResNet-32. * denotes results from code provided by [29] in case the number was not reported in the original paper.

| | CUB Birds 200-2011 | | | |
|---|---|---|---|---|
| *Dimensions* | 3 | 10 | 50 | 100 |
| Mettes *et al.* [29] | 1.0* | 9.5* | 41.5* | **45.6***|
| This paper | **36.6** | **43.5** | **43.5** | **45.6** |

hyperspherical prototypes obtain 1.0% accuracy and we obtain 36.6%. On CIFAR-10, our performance is hardly affected by using few dimensions. The accuracy for our approach in two dimensions is still 91.2% compared to 92.3% in five dimensions and 30.0% in two dimensions for hyperspherical prototypes. The high scores with few dimensions highlights the effectiveness of our approach and is in line with the effectiveness of hyperbolic embeddings in lower dimensionalities [30].

Table 3: **Comparison with varying architecture and prototypes** on CIFAR-100. * denotes results from code provided by [29] in case the number was not reported in the original paper. Our approach is applicable to and effective for any choice of network backbone and choice of prototypes.

| | DenseNet-121 | | | | | ResNet-32 w/ prior knowledge | | | | |
|---|---|---|---|---|---|---|---|---|---|---|
| *Dimensions* | 3 | 5 | 10 | 25 | 50 | 3 | 5 | 10 | 25 | 50 |
| Mettes *et al.* [29] | 11.5* | 45.1* | 60.5 | 69.1 | 71.1* | 11.5 | 37.0 | 57.0 | 64.0 | 65.2* |
| This paper | **66.1** | **69.0** | **71.1** | **72.7** | **71.8** | **52.2** | **56.4** | **62.4** | **65.9** | **68.0** |

To show that the obtained results generalize to other network backbones and choice of prototypes, we report additional results in Table 3 for CIFAR-100. Using a DenseNet-121 architecture improves both our approach and the baseline with similar relative performance improvements with lower dimensionalities. We also investigate the addition of prior semantic knowledge in prototypes as per [29]. With these prototypes, the improvements with few dimensions holds and additionally boost the performance in high dimensions. With 50 dimensions, we obtain an

Table 4: **Comparative evaluation** to softmax cross-entropy, Euclidean prototypes, and hyperspherical prototypes on CIFAR-100 with ResNet-32. Baseline numbers taken from [29].

| | Manifold | Prior knowledge | |
| | | w/o | w/ |
|---|---|---|---|
| Softmax CE | - | 62.1 | - |
| Guerriero *et al.* [17] | $\mathbb{R}^{100}$ | 62.0 | - |
| Mettes *et al.* [29] | $\mathbb{S}^{99}$ | 65.0 | 65.2 |
| This paper | $\mathbb{B}^{50}$ | **65.8** | **68.0** |

accuracy of 68.0%, compared to 65.8% without using prior knowledge. In Table 4, we show additional comparisons to softmax cross-entropy and Euclidean prototypes, which re-iterate the potential of our approach both with and without prior knowledge for prototypes. Overall, we conclude that our approach is applicable and effective for classification regardless of backbone and choice of prototypes.

### 4.3 Comparison to hyperbolic prototypes

**Setup.** The second experiment has shown that Hyperbolic Busemann Learning boosts the classification results compared to its hyperspherical counterpart. In the third experiment, we draw a comparison to the recent hyperbolic action network [24], the only hyperbolic prototype approach to date which is not restricted to few-shot or zero-shot settings. In [24], action classes are embedded as prototypes in the interior of the Poincaré ball based on given hierarchical relations. In contrast, we position prototypes on the ideal boundary of the Poincaré ball, which can be done both with and without prior hierarchical knowledge. To showcase this possibility, we have performed a comparative evaluation on ActivityNet [10] and Mini-Kinetics [48] for the search by action name task. The ActivityNet dataset is a large untrimmed video dataset, which have been trimmed in [24] based on the available temporal annotations. The trimmed dataset contains ~23K videos and 200 classes, with a ~15K split for training and ~8K for validation. Alongside trimming, a modified and more balanced ActivityNet hierarchy with three levels of granularity is provided [24]. The Mini-Kinetics dataset consists of ~83K trimmed videos and 200 classes, with a ~78K split for training and ~5K for validation.We use the hierarchy with three levels of granularity for the Mini-Kinetics as detailed in [24].

We use both the prototypes from prior hierarchical knowledge [24] with entailment cones [15] and knowledge-free prototypes from separation [29]. For our approach, we project the hierarchical prototypes on the ideal boundary through $\ell_2$-normalization. For both datasets, we use the same pre-trained model for feature extraction and three-layer MLP for final classification as [24], with Adam as optimizer, a learning rate of 1e-4, and learning rate drop after 900 iterations. All other hyperparameters are the same as in previous experiments. Following [24], we report three metrics:

Table 5: **Hyperbolic prototypes comparison** on ActivityNet for the search by video name task. When using prototypes from hierarchical knowledge, we score best across the three metrics when using 10 output dimensions. Where [24] is restricted to prototypes from hierarchies, our approach can also employ knowledge-free prototypes, *e.g.,* from separation, to boost the standard mAP metric.

| | **Prototypes from hierarchies** | | | | **Prototypes from separation** | | | |
| *Dimensions* | 2 | 10 | 20 | 50 | 2 | 10 | 20 | 50 |
|---|---|---|---|---|---|---|---|---|
| Long *et al.* [24] | | | | | | | | |
| mAP | 26.9 | 68.3 | 66.9 | 67.1 | - | - | - | - |
| s-mAP | 72.0 | 93.8 | 92.7 | 93.1 | - | - | - | - |
| c-mAP | 91.9 | 97.7 | 97.2 | 97.5 | - | - | - | - |
| This paper | | | | | | | | |
| mAP | 40.8 | **70.6** | 66.3 | 65.6 | 44.1 | 78.0 | **79.4** | 78.5 |
| s-mAP | 79.2 | **95.1** | 94.5 | 94.7 | 48.9 | **88.6** | 86.2 | 86.7 |
| c-mAP | 92.9 | 98.0 | 97.8 | **98.2** | 60.8 | 84.0 | 90.6 | **91.7** |

Table 6: **Hyperbolic prototypes comparison** on mini-Kinetics for the search by video name task. When comparing the models trained on the prototypes from hierarchies, our model outperforms when using low dimension embeddings.

| | **Prototypes from hierarchies** | | | | **Prototypes from separation** | | | |
|---|---|---|---|---|---|---|---|---|
| *Dimensions* | 2 | 10 | 20 | 50 | 2 | 10 | 20 | 50 |
| Long *et al.* [24] | | | | | | | | |
| mAP | 28.8 | 64.4 | 63.5 | 64.5 | - | - | - | - |
| s-mAP | 79.0 | 94.2 | 94.1 | 94.5 | - | - | - | - |
| c-mAP | 91.0 | 96.5 | 96.3 | 96.5 | - | - | - | - |
| This paper | | | | | | | | |
| mAP | 51.7 | **67.5** | 65.0 | 64.0 | 56.4 | 70.7 | 71.6 | **71.7** |
| s-mAP | 88.3 | **95.6** | 95.1 | 95.0 | 62.0 | 76.8 | 78.8 | **80.0** |
| c-mAP | 94.1 | **97.1** | **97.1** | 97.0 | 71.0 | 82.5 | 84.5 | **85.6** |

mean Average Precision (mAP), sibling-mAP (s-mAP), which counts a prediction as correct if it is within 2-hops away in the hierarchy, and cousin-mAP (c-mAP), which does the same for 4-hops.

**Results.** The comparative evaluation on ActivityNet and Mini-Kinetics is shown in Table 5 and Table 6. When hierarchical knowledge is available, we only position the leaf nodes on the boundary and the internal nodes of the hierarchy will be positioned inside the Poincaré ball. Using the prototypes from hierarchical knowledge, we find that both our approach and the baseline favor low-dimensional embeddings. Our highest score for the mAP metric on Activity-Net (70.6%) outperforms the baseline (68.3%) and the same holds for sibling-mAP (95.1% versus 93.8%) and cousin-mAP (98.2% versus 97.5%). The same holds on Mini-Kinetics. These results highlight our ability to employ prototypes with prior knowledge. We note that the baseline performs additional tuning for the curvature in hyperbolic space, while we only use a standard curvature of one. A unique benefit of our approach is the ability to employ hyperbolic spaces without requiring hierarchical priors for prototypes. We find that positioning prototypes solely based on data-independent separation provides a boost in mAP, from 68.3% by [24] to 79.4%. In contrast, the hierarchical metrics score lower with the prototypes from separation. This is a direct result of the lack of hierarchical knowledge during prototype positioning. An explanation for the better performance of hierarchical prototypes over separation prototypes for s-mAP and c-mAP compared to standard mAP comes from how the prototypes are constructed. For the hierarchical prototypes, classes with the same parent will be pulled together, while other classes will be pushed away. This boosts the s-MAP and c-mAP since these metrics allow for confusion with hierarchically related classes. However, the standard mAP considers all other classes as negatives. In that case, separation prototypes are preferred as they push all classes away during prototype construction. Overall, we conclude that Hyperbolic Busemann Learning provides both an effective and general approach for prototype-based learning with hyperbolic output spaces.

## 5 Conclusions

In this paper, we introduce Hyperbolic Busemann Learning with ideal prototypes. Where existing work on prototype-based learning in hyperbolic space requires prior knowledge to operate, we propose to place prototypes at the ideal boundary of the Poincaré ball. This enables a prototype positioning without the need for prior knowledge. We introduce the penalized Busemann loss to optimize examples with respect to ideal prototypes. We provide both theoretical and empirical support for the idea of ideal prototypes and the proposed loss. We prove its equivalence to logistic regression in dimension one and show on three datasets that our approach outperforms recent hyperspherical and hyperbolic prototype approaches.

**Broader impact.** Our approach allows for both knowledge-based and knowledge-free prototypes and this choice matters depending on its broader utilization. Knowledge-free prototypes make fewer errors but the remaining ones are less hierarchically consistent. Hence critical applications such as in the medical domain benefit from the former, while search-based applications benefit form the latter.

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

## A   Full derivation of Busemann function in Poincaré model

In the Poincaré model, the unit-speed geodesic $\gamma_p(t)$ from the origin towards the ideal point $p$ is given by
$$\gamma_p(t) = p \tanh(t/2).$$
Inserting into (6) and using the hyperbolic distance as given in (3) we obtain the following representation of the Busemann function:
$$b_p(z) = \lim_{t \to \infty} \Big( \text{arcosh } (1 + x(t)) - t \Big),$$
where
$$x(t) = 2 \frac{\|p \tanh(t/2) - z\|^2}{(1 - \tanh(t/2)^2)(1 - \|z\|^2)}.$$
Using that $\tanh(t/2) = (e^t - 1)/(e^t + 1)$, it follows that
$$\lim_{t \to \infty} e^{-t} x(t) = \frac{1}{2} \frac{\|p - z\|^2}{1 - \|z\|^2}.$$
In particular, this shows that $x(t)$ grows to infinity as $t \to \infty$. Using the representation of the inverse hyperbolic cosine in terms of the logarithm, we obtain (with Landau's small-$o$ notation)
$$\text{arcosh } (1 + x(t)) = \log \Big( 1 + x(t) + \sqrt{x(t)^2 + 2x(t)} \Big) = \log \Big( 2x(t) + o(x(t)) \Big),$$
as $x(t) \to \infty$. Thus, it follows that
$$\begin{aligned}
b_p(z) &= \lim_{t \to \infty} \Big\{ \log \Big( 2x(t) + o(x(t)) \Big) - t \Big\} = \lim_{t \to \infty} \log \Big( 2e^{-t} x(t) + e^{-t} o(x(t)) \Big) = \\
&= \log \Big( 2 \lim_{t \to \infty} e^{-t} x(t) + 0 \Big) = \log \frac{\|p - z\|^2}{1 - \|z\|^2},
\end{aligned}$$
as claimed in (7).

