# OpenReview forum: "Hyperbolic Busemann Learning with Ideal Prototypes"
_NeurIPS.cc/2021/Conference — NeurIPS 2021 Poster_

### Official Review · Reviewer_2AsY · 2021-07-15

**Rating:** 6
**Confidence:** 3

**Summary:**

Paper proposes an hyperbolic learning method based on ideal prototypes. The originality of the paper lies in the fact that there is no need to include prior information on the hierarchy to position the prototypes. There are located at the boundary of the Poincaré ball, and their position is computed by using algorithm defined in ref [28]. As points located close to the boundary have an infinite distance to the origin, authors propose to use a penalized busemann loss that can be defined in the hyperbolic space. The penalty term allows defining how far the embeddings can be from the origin. Paper also shows that, in 1d, it corresponds to logistic regression.
Experiments are provided, considering CIFAR10 and 100 datasets, on classification tasks. Results show that the proposed method provides better results than the baselines when the dimension is low. When no information on the structure of the prototypes is given, the mAP indicator is better than when the prototypes are inferred from the data, while hierarchical metrics such as s-mAP and c-mAP show that the learnt prototypes perform better when privileged information is provided.

**Limitations And Societal Impact:**

yes

**Main Review:**

The paper is very clear and well written. The introduction of the busemann loss is smart and allows defining an elegant loss, with a link with logistic regression in 1d.
My main concern is about the significance of the paper: the algorithm is shown to be effective in low dimensional embedding spaces (as often noticed in the litterature) and the performances come close to the spherical version of [28] when the dimension increases. The paper should emphasize some use cases on which it is important to have such low dimensional spaces (when the classification performance is sought, why considering low dimensional spaces that degrades the performances?). In Table 4, one can notice that ref [23] gives comparable performances when the dimension reaches 50 and one may wonder how the proposed method behaves in larger spaces (one can see that the performances seem to degrade when the dimension increases?).
In addition, some results in Table 2 seem surprising. While s-map and c-map perform better when hierarchical prototypes rather than separation prototypes are used, it is not the case for the mAP metrics. It would be interesting to give an explanation.

Minor comments:
- Code was not made available to re-run the experiments
- Table 4: ref [23] instead of [28]
- it would strengthen the conclusion if an other dataset could be considered (not only CIFAR).
- results are only given for lower dimensional spaces (e.g. 50 instead of 99 or 100) in table 3: why not considering the same dimension?


**Time Spent Reviewing:**

4

---

> ### Author Response · Authors · 2021-08-10
> **Response to Reviewer 2AsY**
>
> We thank the reviewer for the positive feedback on writing, the introduced loss, and the link to logistic regression. Below, we have addressed the raised questions and suggestions.
>
> ### Main concern
> Indeed, a lot of our empirical benefit is when using low-dimensional output spaces. This is a common goal in hyperbolic deep learning [18, 22, 23] and has several practical use cases and benefits. We list two important use cases below:
>
> - **Compactness**: Hyperbolic Busemann Learning enables us to embed high-dimensional inputs in low-dimensional output spaces. This ability has several practical benefits in applications such as data transmission, compression, and storage.
>
> - **Interpretability**: When training a model on CIFAR10 with only two output dimensions, we maintain over 90% of performance compared to the 10-dimensional case. By using 2-dimensional output space in the paper, we are able to gain additional interpretability, including the insights into the penalty term (Figure 4) and the confidence interpretation in the output space (Figure 5).
> The ability and potential of learning with low-dimensional output spaces are the reason why we focus on fewer dimensions in the paper. Following the suggestion by the reviewer, we will include the potential use cases of low-dimensional output spaces in the paper.
>
> ### Performance of hierarchical prototypes versus separation prototypes
> An explanation for the better performance of hierarchical prototypes over separation prototypes for s-mAP and c-mAP compared to standard mAP comes from how the prototypes were constructed. For the hierarchical prototypes, classes with the same parent will be pulled together, while other classes will be pushed away. This boosts the s-MAP and c-mAP since these metrics allow for confusion with hierarchically related classes. The standard mAP however considers all other classes as negatives. In that case, separation prototypes are preferred as they push all classes away during prototype construction. We will include this discussion in Section 4.3.
>
> ### Considering other datasets
> Based on the suggestion of the reviewer, we added extra experiments on the CUB Birds 200-2011 dataset [a], which consists of 200 bird categories. We report the results of our model compared to [28] using 3, 10, 50, and 100 output dimensions in the table below.
>
> | | 3 | 10 | 50 | 100 |
> |---|---|---|---|---|
> | Mettes et al. [28] | 1.0 | 9.5 | 41.5 | **45.6** |
> | This paper | **36.6** | **43.5** | **43.5** | **45.6** |
>
> We have also performed additional experiments on the Mini-Kinetics dataset [b], which contains 200 classes, for the search by video name task. We report the results akin to Table 4. All the hyperparameters are the same as reported in Section 4.3.
>
> | prototypes from hierarchies|| 2 | 10 | 20 | 50 || prototypes from separation| 2 | 10 | 20 | 50 |
> |---|--|---|---|---|---||---|---|---|---|---|
> | Long et al. [23]| mAP| 28.8 | 64.4 | 63.5 | 64.5 ||  | - | - | - | - |
> ||                          s-mAP| 79.0 | 94.2 | 94.1 | 94.5 ||  | - | - | - | - |
> | | c-mAP| 91.0 | 96.5 | 96.3 | 96.5 ||  | - | - | - | - |
> |This Paper | mAP | 51.7 | **67.5** | 65.0 | 64.0 ||  | 56.4 | 70.7 | 71.6 | **71.7** |
> | | s-mAP| 88.3 | **95.6** | 95.1 | 95.0 ||  | 62.0 | 76.8 | 78.8 | **80.0** |
> | | c-mAP| 94.1 | **97.1** | **97.1** | 97.0 ||  | 71.0 | 82.5 | 84.5 | **85.6** |
>
> We will include the results on Mini-Kinetics in the supplementary materials.
>
> ### Minor comments
> We apologize for not emphasizing the availability of the code. The code is available in the supplementary materials. This code allows for re-running Tables 1, 2, 3. We will make the GitHub project public after the author notification date to re-run all the experiments. We thank the author for pointing out the incorrect citation in Table 4. We will change the paper accordingly.
>
> [a] C.Wah, S.Branson, P.Welinder, P.Perona, and S.Belongie. The Caltech-UCSD Birds-200-2011 Dataset. Technical Report CNS-TR-2011-001, California Institute of Technology, 2011.
>
> [b] Saining Xie, Chen Sun, Jonathan Huang, Zhuowen Tu, and Kevin Murphy. Rethinking spatiotemporal feature learning: Speed-accuracy trade-offs in video classification. In ECCV, 2018

---

> > ### Comment · Reviewer_2AsY · 2021-08-30
> > **Thanks for your response**
> >
> > I would like to thank you for providing a clear response to my comments. It addresses my main concerns and as such, I raised my score from 5 to 6.
> > I believe that the results on the additional dataset gives an additional experimental strength to the paper.
> > I suggest to add a discussion on the interest of low-dimensional spaces and why one sometimes observe a degradation (non significant?) of the performances when increasing the dimensionality of the embedding space.

---

> > > ### Author Response · Authors · 2021-08-30
> > > **Response to Reviewer 2AsY**
> > >
> > > We are glad that the main concerns have been addressed. Following the reviewer's suggestion, we will include the discussion on the potential of low-dimensional spaces and the behaviour when increasing the dimensionality of the embedding space to the paper.

---

### Official Review · Reviewer_HYUM · 2021-07-17

**Rating:** 8
**Confidence:** 4

**Summary:**

This paper introduces a novel model which uses prototypes placed "at infinity" in hyperbolic space for classification. Specifically, ideal prototypes are placed on the surface of the sphere, and the distance from a point inside the sphere the this idealized point is calculated using the Busemann function, which defines a distance to points at infinity using a limit. Thus, the embedded points representing instances can be interpreted as points in hyperbolic space, and prototypes as points at infinity. The authors perform a number of rigorous evaluations and ablation analysis, demonstrating their model performs well in a number of classification settings.

**Limitations And Societal Impact:**

Some limitations are touched on. While the paper does not oversell it's methods, it is also not particularly self-critical, and some of the questions I raise above do suggest various limitations of the model which might be explored further.

The work is of a theoretical nature, and does not present any direct negative societal impact.

**Main Review:**

### Overall

**Originality:** This paper is reasonably novel, leveraging the Busemann loss function to provide a hyperbolic analog for prototype classification methods.

**Quality:** The submission is technically sound. The approach is reasonable, claims made are proven directly in the paper and appear to be correct.

**Clarity:** The paper is very clearly written and well organized.

**Significance:** The results are significant, extending tools and methods researchers are interested in to the hyperbolic setting.

---
### Specific Comments / Questions

- (Comment) I was initially prepared to discard the connection to logistic regression in one dimension, as I assumed this would simply be an uninteresting degenerate case, however the connection the authors drew to higher-dimensional models with prototypes being an alternative to multinomial logistic regression did provide interesting motivation.
- The ablation for the penalized Busemann loss suggest that a larger number of classes will be much more sensitive to the weight of the penalty term. This seems as though it may be inherent to the nature of hyperbolic space, for which volumes of spheres increase exponentially with respect to the radius and thus (in some sense) the penalty term is severely limiting the capacity of the model. Some theoretical exploration or additional experiments on datasets with larger number of labels would provide more confidence that this is not an issue.
- Much of the claimed benefit for hyperbolic embeddings comes from their inductive bias toward modeling hierarchical data. Placing the prototypes for hierarchical data at the boundary would seemingly discard this essential property. In short, it seems as though you would not *want* to use idealized prototypes, placed on the boundary of the sphere, in the case where the prototypes themselves conform to a hierarchy. I would appreciate it if the authors could clarify this concern.

**Time Spent Reviewing:**

5

---

> ### Author Response · Authors · 2021-08-10
> **Response to Reviewer HYUM**
>
> We thank the reviewer for their positive comments on novelty, writing, and significance of results. We are glad to see that the connection to logistic regression provides an interesting motivation. Below, we have addressed the raised questions and suggestions.
>
> ### Penalized Busemann loss and penalty term sensitivity
> Based on the suggestion of the reviewer, we have performed additional experiments on the CUB Birds 200-2011 dataset [a] and Mini-Kinetics [b], both of which contain 200 categories. For CUB Birds, we keep the main hyperparameters the same as for the CIFAR experiments and use a weight decay of 0.0001, 2110 epochs, and a slope of 0.05. We report the results of our model compared to [28] using 3, 10, 50, and 100 output dimensions in the table below. All experiments are performed on ResNet32, akin to Table 1.
>
> | | 3 | 10 | 50 | 100 |
> |---|---|---|---|---|
> | Mettes et al. [28] | 1.0 | 9.5 | 41.5 | **45.6** |
> | This paper | **36.6** | **43.5** | **43.5** | **45.6** |
>
> Similar to the CIFAR-10 and CIFAR-100 experiments, our improvements over [28] grow as dimensionality shrinks. This result indicates that the empirical results also hold for 200 categories. As the reviewer pointed out, a lower slope for the penalty term is preferred when using more classes. Using the penalty term remains important and removing it also hurts performance with 200 classes.
>
> The results for Mini-Kinetics are shown in the table below, using the same hyperparameters as reported in Section 4.3.
>
> | prototypes from hierarchies|| 2 | 10 | 20 | 50 || prototypes from separation| 2 | 10 | 20 | 50 |
> |---|--|---|---|---|---||---|---|---|---|---|
> | Long et al. [23]| mAP| 28.8 | 64.4 | 63.5 | 64.5 ||  | - | - | - | - |
> ||                          s-mAP| 79.0 | 94.2 | 94.1 | 94.5 ||  | - | - | - | - |
> | | c-mAP| 91.0 | 96.5 | 96.3 | 96.5 ||  | - | - | - | - |
> |This Paper | mAP | 51.7 | **67.5** | 65.0 | 64.0 ||  | 56.4 | 70.7 | 71.6 | **71.7** |
> | | s-mAP| 88.3 | **95.6** | 95.1 | 95.0 ||  | 62.0 | 76.8 | 78.8 | **80.0** |
> | | c-mAP| 94.1 | **97.1** | **97.1** | 97.0 ||  | 71.0 | 82.5 | 84.5 | **85.6** |
>
> We additionally note that the form of the penalty term as outlined in Theorem 1 is independent of the number of classes. We will insert the experiments on CUB Birds 200-2011 in the paper and on Mini-Kinetics in the supplementary materials.
>
> ### Hierarchical prior in hyperbolic ideal prototypes
> We apologize for the missing details on how to use hierarchical knowledge in our approach. When hierarchical data is available, we only position the leaf nodes on the boundary. The internal node of the hierarchy will be positioned inside the Poincaré ball. As a result, we can still benefit from the given hierarchical structure of the labels, with its empirical benefits on hierarchical metrics as shown in Table 4. We will include the details in Section 4.3.
>
> [a] C.Wah, S.Branson, P.Welinder, P.Perona, and S.Belongie. The Caltech-UCSDBirds-200-2011 Dataset. Technical Report CNS-TR-2011-001, California Institute of Technology, 2011.
>
> [b] Saining Xie, Chen Sun, Jonathan Huang, Zhuowen Tu, and Kevin Murphy. Rethinking spatiotemporal feature learning: Speed-accuracy trade-offs in video classification. In ECCV, 2018

---

> > ### Comment · Reviewer_HYUM · 2021-09-02
> > **Appreciate the additional experiments and clarifications**
> >
> > Thank you for performing additional experiments with a larger number of categories.
> >
> > There are a variety of ways in which the experimental section of this work could be strengthened, in particular:
> > 1. Rigorous tuning of hyperparameters for each experiment / dimension (preferably via Bayesian optimization, though random would be reasonable also).
> > 2. Including the standard error with respect to different random seeds.
> >
> > Regarding the hierarchical prior: thanks for the clarification. Have you evaluated how the performance improvement is related to the depth of the nodes? Here, I am wondering if the primary benefit comes from improved classification of leaf nodes, for example.
> >
> > I agree with the other reviewer's suggestion to include a discussion of the relevance for low-dimensional embeddings; indeed, if one is principally concerned in performance then standard dot product baselines are often the way to go. I am aware that it is characteristic of hyperbolic embeddings to principally outperform in lower dimensions, and for this performance penalty to saturate or even degrade as dimension is increased, so providing clear motivation for why one would be interested in these low-dimensional settings would add great value to the future readers of this work.
> >
> > I feel my overall assessment of this paper is accurate, and am happy to keep my rating as it stands currently.

---

> > > ### Author Response · Authors · 2021-09-03
> > > **Response to Reviewer HYUM**
> > >
> > > We thank the reviewer for the suggestion. We will include the hyperparameter tuning in the supplementary materials and the standard errors to the paper, as well as the low-dimensionality discussion suggested by reviewer 2AsY.
> > >
> > > Regarding the hierarchical prior, the mAP refers to performance over all leaf nodes, while s-mAP and c-mAP refer to performance over parent and grandparent nodes. The improvements with the use of hierarchical priors is on the s-mAP and c-mAP, i.e. the improvements come from better results for non-leaf nodes. We will clarify in the paper.

---

### Official Review · Reviewer_28qA · 2021-07-17

**Rating:** 7
**Confidence:** 3

**Summary:**

The paper proposes Hyperbolic Busemann Learning for hyperbolic prototypes. Specifically, the prototypes are positioned on the ideal boundary of the Poincare ball model and the penalised Busemann loss is used to learn the projections of the input data to the Poincare ball model. The paper also provides a theoretical link between the proposed hyperbolic prototype approach and logistic regression.

**Limitations And Societal Impact:**

Yes.

**Main Review:**

The proposed hyperbolic prototype network is novel and well motivated. The usage of Busemann function to compute the distance between the ideal boundary point and the point in the Poincare ball model is practical. The empirical evaluation demonstrates the effectiveness of the proposed approach.

Some questions:
1. The derivation of the Busemann function in the Poincare ball model Eq. (7) is missing. How to obtain Eq. (7) from Eqs. (3) and (6) is not trivial. The derivation process or the corresponding reference should be given.
2. I have confusion on the penalty term. As mentioned in Section 3.2, the penalty term is the same for all prototypes, thus it is equivalent with minimizing the Busemann function of the prototypes directly. Then why does the penalty term affect the test accuracy in Section 4.1?
3. Another confusion is in Line 145-146. It said that the Busemann function is a decreasing function of the cosine similarity, but there is no such concept of 'cosine similarity' in the hyperbolic space.

Some revision suggestions (which do not influence the overall evaluation of the paper):
1. Although hyperbolic space has shown many improvements, it still has some limitations, the 'arbitrary data' in the abstract may be inappropriate and too absolute.
2. In Line 167, there is an extra comma.

**Time Spent Reviewing:**

3.5

---

> ### Author Response · Authors · 2021-08-10
> **Response to reviewer 28qA**
>
> We thank the reviewer for their positive comments regarding the novelty of the approach, the motivation, and the empirical effectiveness. Below, we have addressed the raised questions and suggestions.
> ### Full derivation of the Busemann function
> We thank the reviewer for the suggestion to provide the full derivation of the Busemann function in the paper. In the Poincaré model, the unit-speed geodesic $\gamma_p(t)$ from the origin towards the ideal point $p$ is given by
>
> $$\gamma_p(t) = p \tanh(t/2).$$
>
> Inserting into (6) and using the hyperbolic distance as given in (3), we obtain the following representation of the Busemann function:
>
> $$
> b_p(z) = \lim_{t \rightarrow \infty} \Big(\mathrm{arcosh}\left(1 + x(t)\right) - t\Big),
> $$
>
> where
>
> $$x(t) = 2 \frac{\left\Vert p \tanh(t/2) - z\right\Vert^2}{(1 - \tanh(t/2)^2)(1 - \left\Vert z\right\Vert^2)}.$$
>
> Given that $\tanh(t/2) = (e^t - 1)/(e^t + 1)$, it follows that
>
> $$\lim_{t \to \infty} e^{-t}x(t) = \frac{1}{2}\frac{\Vert p - z\Vert^2}{1 - \Vert z\Vert^2}.$$
>
> In particular, this shows that $x(t)$ grows to infinity as $t \to \infty$. Using the representation of the inverse hyperbolic cosine in terms of the logarithm, we obtain (with Landau's small-$o$ notation)
>
> $$\mathrm{arcosh}\,\left(1 + x(t)\right) = \log\left(1 + x(t) + \sqrt{x(t)^2 + 2x(t)}\right) = \log \Big(2x(t) + o(x(t))\Big),$$
>
> as $x(t) \to \infty$. Thus, it follows that
>
> $$
> b_p(z) = \lim_{t \rightarrow \infty}\Big({\log \Big(2x(t) + o(x(t))\Big) - t\}\Big) = \lim_{t \rightarrow \infty} \log \Big(2 e^{-t} x(t) + e^{-t}o(x(t))\Big) = $$ $$= \log \left(2 \lim_{t \to \infty} e^{-t} x(t) + 0\right) = \log \frac{\Vert p - z\Vert^2}{1 - \Vert z\Vert^2},
> $$
>
> as claimed in (7). We will incorporate the full derivation.
>
> ### The role of the penalty term in training and testing
> Directly minimizing the Busemann function of the prototypes only holds for testing. Since the penalty term is the same for all prototypes, we can ignore it when assigning labels to examples. The penalty term is however influential during training, as it balances confidence and inter-class separation. As a result, different settings for the penalty term influence the overall test accuracy. We will make the effect of the penalty term more clear in Section 3.2.
>
> ### Busemann function and angles
> We agree that the term cosine similarity in the context of lines 145-146 is confusing. We will replace the cosine similarity with "angle between prototype and network output".
>
> ### Revision Suggestions
> We will remove the word arbitrary from the abstract and the extra comma from line 167. Thank you.

---

> > ### Comment · Reviewer_28qA · 2021-08-31
> > **Thank you for your response**
> >
> > I appreciate the response of the authors, which addressed my confusion and questions. I update my score from 6 to 7.
> >
> > For the Busemann function and angles, I would suggest the authors to clearly define the angle in the hyperbolic space after replacing 'the cosine similarity' with 'angle between prototype and network output'.

---

> > > ### Author Response · Authors · 2021-08-31
> > > **Response to reviewer 28qA**
> > >
> > > We thank the reviewer for their guidance and we will add the appropriate angle definition with reference to the paper.

---

### Decision · Program_Chairs · 2021-09-27

**Decision:**

Accept (Poster)

**Comment:**

The paper proposes a new approach for classification in hyperbolic space by combining ideal prototypes at the boundary of hyperbolic space and a penalized Busemann loss. The paper is well written, clear, and relevant to the NeurIPS community. All reviewers and the AC support acceptance, especially due to the novel and interesting ideas that advance research on hyperbolic neural networks and their applications, as well as the promising empirical results of the method. Reviewers highlighted also that the approach is technically sound, claims are supported in the paper, and the good empirical evaluation. Furthermore, questions and concerns related to the derivation of the approach, motivation, and significance of results where resolved after rebuttal. When preparing the camera ready version, please incorporate the overall feedback and suggestions of reviewers into the new revision.